# Rutin Promotes Pancreatic Cancer Cell Apoptosis by Upregulating miRNA-877-3p Expression

**DOI:** 10.3390/molecules27072293

**Published:** 2022-03-31

**Authors:** Mingxing Huo, Aowen Xia, Wenwen Cheng, Mengjie Zhou, Jiankang Wang, Tiantian Shi, Cifeng Cai, Wenqi Jin, Meiliang Zhou, Yueling Liao, Zhiyong Liao

**Affiliations:** 1Zhejiang Provincial Key Laboratory for Water Environment and Marine Biological Resources Protection, College of Life and Environmental Science, Wenzhou University, Wenzhou 325000, China; 194511382328@stu.wzu.edu.cn (M.H.); awxia666@163.com (A.X.); chengwenwen@zju.edu.cn (W.C.); 194511381305@stu.wzu.edu.cn (M.Z.); 194511381309@stu.wzu.edu.cn (J.W.); sttwzu@163.com (T.S.); caicifeng1988@sina.com (C.C.); 2Department of Anorectal Surgery, Shanghai Municipal Hospital of Traditional Chinese Medicine, Shanghai University of Traditional Chinese Medicine, Shanghai 200071, China; jwq0910@126.com; 3Institute of Crop Sciences, Chinese Academy of Agricultural Sciences, Beijing 100081, China; zhoumeiliang@caas.cn

**Keywords:** rutin, pancreatic cancer, miRNA-877-3p, Bcl-2

## Abstract

(1) Background: pancreatic cancer is one of the most serious cancers due to its rapid and inevitable fatality, which has been proved very difficult to treat, compared with many other common cancers. Thus, developing an effective therapeutic strategy, especially searching for potential drugs, is the focus of current research. The exact mechanism of rutin in pancreatic cancer remains unknown. (2) Method: three pancreatic cancer cell lines were used to study the anti-pancreatic cancer effect of rutin. The potent anti-proliferative, anti-migration and pro-apoptotic properties of rutin were uncovered by cell viability, a wound-healing migration assay, and a cell apoptosis assay. High-throughput sequencing technology was used to detect the change of miRNAs expression. Immunoblotting analysis was used to detect the expression of apoptotic proteins. (3) Results: CCK-8 and EDU assays revealed that rutin significantly inhibited pancreatic cancer cells’ proliferation (*p* < 0.05). A wound-healing assay showed that rutin significantly suppressed pancreatic cancer cells’ migration (*p* < 0.05). A flow cytometric assay showed that rutin could promote pancreatic cancer cells’ apoptosis. Intriguingly, rutin significantly upregulated miR-877-3p expression to repress the transcription of Bcl-2 and to induce pancreatic cancer cell apoptosis. Accordingly, rutin and miR-877-3p mimics could promote apoptotic protein expression. (4) Conclusions: our findings indicate that rutin plays an important role in anti-pancreatic cancer effects through a rutin-miR-877-3p-Bcl-2 axis and suggests a potential therapeutic strategy for pancreatic cancer.

## 1. Introduction

Pancreatic cancer (PC) is one of the most serious cancers due to its high metastasis and mortality rate [1]. Currently, surgical excision is the most effective treatment option for PC, but the 5-year survival rate is only 6–7% [2]. Therefore, it is urgent to find new effective drugs or new potential targets for PC therapy. Dysregulations of the JAK-STAT signal pathway, the NF-κB signaling pathway or the Notch signal pathway were reported to be extensively involved in the development of pancreatic cancer, among which pathways some signaling molecules have become promising potential therapeutic targets for PC therapy [3,4,5,6]. Increasing evidence has shown that natural products may play a promising role in the development of novel chemotherapeutics for PC therapy [7]. Imran et al. found that plant-derived luteolin was beneficial in the treatment of pancreatic cancer by inducing cell apoptosis and cell-cycle arrest and inhibiting protein phosphorylation and signal transduction [8]. Another study also indicated that luteolin induced apoptosis of pancreatic cancer cells in vivo by inhibiting KRAS-GSK-3β-NF-κB signaling pathways, accompanied by cytochrome C release, caspase 3 activation, and Bcl-2/Bax ratio decreases [9]. Despite the importance of natural products in PC therapy, the mechanisms for this remain unknown.

Natural flavonoids are a major resource of potential drugs because of their unique biological activity and their relative non-toxicity. Studies have found that flavonoids protect cells against cellular injury to reduce the risk of cancers [10]. Veeriah et al. described that the flavonoid extracted from apples could inhibit colon cancer by regulating the expression of GSTP1, GSSTT2, MGST2, CHST5, CHST6 and CHST7 [11]. Elkady et al. reported that the medicinal herb Nigella sativa flavonoid can inhibit proliferation and induce apoptosis in MCF-7 cells [12]. Rutin is a major flavonoid that is found in abundance in plants such as passionflower, tea, apple, and especially in Tartary buckwheat [13]. Rutin has attracted increasing attention because of its obvious antioxidant, anti-tumor and anti-inflammatory effects [14,15,16]. It has been shown that rutin modulated Janus kinase/signaling (MAPK), NF-κB, PI3K/Akt and Wnt/β-catenin signaling in cancer cells. Moreover, rutin modified intercellular signaling pathways and intermediate cross-talk pathways, including miRNAs, IncRNAs and mRNAs [17]. Chen et al. found that rutin induced G2/M cell cycle arrest and promoted apoptosis in human neuroblastoma cell lines [18]. Rutin effectively ameliorated the expression of NF-κB inflammatory pathway members, including NF-κB, IκB kinase (IKK)-α, and IKK-β in HT29 colon cancer cells [19]. In addition, rutin ceased the cell cycle at the sub-G1 phase by regulating the miRNAs-lncRNAs-mRNAs-TFs network [20]. Recently, the regulatory effect of rutin on microRNA (miRNA) expression has been widely discussed. A study indicated that rutin could significantly ameliorate pirarubicin-induced oxidative damage and cardiomyocyte apoptosis in rats by regulating miR-22-5p expression [21]. Another study reported that rutin could inhibit the growth and metastasis of 4T1 mouse breast cancer cells in a dose-dependent manner by increasing the miR-129-1-3p expression level [22]. However, the regulatory effect of rutin on miRNA is unknown in pancreatic cancer.

As we know, miRNAs are newly discovered candidates for tumor therapy [23,24], which are involved in many biological processes, such as cell proliferation, differentiation, migration, and apoptosis [25]. miR-877-3p is a potential tumor suppressor in cancers [26]. Li et al. found that miR-877-3p overexpression could suppress bladder cancer growth by activating the expression of p16 [26]. Another study indicated that miR-877 may act as a tumor suppressor, inhibiting the proliferation of hepatocellular carcinoma cells [27]. Zhou et al. also reported that miR-877-3p could act as a suppressor of osteosarcoma cell proliferation and exert an anti-angiogenesis effect on osteosarcoma microenvironment [28]. However, the role of miR-877-3p in pancreatic cancer remains unknown. In this study, we found that rutin treatment strongly inhibited the proliferation and migration of pancreatic cancer cells and enhanced pancreatic cancer cell apoptosis. Intriguingly, rutin significantly upregulated miR-877-3p expression, which played an important role in the proliferation, migration, and apoptosis of pancreatic cancer cells. Additionally, miR-877-3p negatively regulated the expression of the anti-apoptotic protein Bcl-2 in PANC-1. Our findings thus provided novel insights into the role of rutin in anti-pancreatic tumor treatment via the rutin-miR-877-3p-Bcl-2 axis and a promising therapeutic strategy for PC.

## 2. Results

### 2.1. Rutin Suppresses Pancreatic Cancer Cells’ Proliferation and Migration In Vitro

We prepared rutin from Tartary buckwheat. It was shown in Figure 1 that the purity of rutin prepared by us reached more than 96%. To investigate the role of rutin in pancreatic cancer cell proliferation, PANC-1 cells were treated with rutin at different concentrations in vitro (5–40 μg/mL). The results of the CCK-8 assay showed that rutin inhibited the growth of PANC-1 cells in a dose-dependent manner compared with the control group, and that 5 μg/mL of rutin could inhibit the growth of PANC-1 cells significantly (Figure 2A). An EdU assay also showed that rutin inhibited PANC-1 cell proliferation at the concentration of 5 μg/mL and 10 μg/mL (Figure 2B). Further, we used concentrations of 5 μg/mL and 10 μg/mL of rutin to explore its effect on the migration of PANC-1 cells. The wound-healing assay showed that the migration of PANC-1 cells was significantly inhibited after the administration of rutin for 48 h (Figure 2C). These findings also existed in SW1990 cells and MIA PaCa-2 cells (Appendix A). In addition, the immunoblotting assay showed that rutin inhibited the expression of the migration-related protein MMP-9 (Figure 2D).

### 2.2. Rutin Promotes Pancreatic Cancer Cells Apoptosis

Next, we investigated the role of rutin in PANC-1 apoptosis using the TUNEL assay, which is used to detect DNA fragmentation in the last phase of apoptosis. After rutin treatment, cell apoptosis was seen to increase, compared with the untreated cells (Figure 3A). Annexin V staining also revealed that the percentage of cells undergoing apoptosis increased after the administration of rutin compared with the control group of PANC-1 cells (Figure 3B). Similarly, rutin also promoted the apoptosis of SW1990 cells and MIA PaCa-2 cells (Appendix A). Moreover, the expression of apoptosis-related proteins (Bax, caspase-3, cleaved-caspase 8, and cleaved-caspase 9) were detected by immunoblotting (Figure 3C). Altogether, our results demonstrate that rutin induces apoptosis in human PC cells.

### 2.3. Rutin Upregulates the Expression of miR-877-3p in Pancreatic Cancer Cells

To explore the molecular mechanism of rutin on human pancreatic cancer, we used high-throughput sequencing technology performed by Beijing Biomarker Technologies Co. Ltd. to detect the change in miRNAs expression with rutin treatment, and the results showed that the expression of miRNA-877-3p was significantly upregulated after rutin treatment (Figure 4A,B). After this, RT-qPCR was used to verify the relationship between rutin and miRNA-877-3p. Consistent with the sequencing results, after rutin treatment, the expression level of miRNA-877-3p was significantly increased in PANC-1 cells (Figure 4C).

### 2.4. Effect of miR-877-3p on the Viability, Migration, and Apoptosis of Pancreatic Cancer Cells

To identify and validate the effect of miRNA-877-3p on cellular behavior, PANC-1 cells were transfected with miRNA-877-3p mimics, miRNA-877-3p inhibitor, and their corresponding control oligonucleotides (miR-NC). After the respective transfection of the miR-877-3p mimics and miR-877-3p inhibitor, the expression of miR-877-3p was upregulated and downregulated with statistical significance in PANC-1 cells (Figure 5A). The CCK-8 and EdU assays exhibited that the miRNA-877-3p mimics significantly suppressed cell proliferation in PANC-1 cells, whereas the miRNA-877-3p inhibitors enhanced the proliferation of PANC-1 cells (Figure 5B,C). Similarly, the miRNA-877-3p mimics also inhibited the proliferation of SW1990 cells and MIA PaCa-2 cells (Appendix A). The wound-healing assay revealed that the PANC-1, SW1990 and MIA PaCa-2 cells with miRNA-877-3p treatment underwent a significant reduction in cell motility, whereas the inhibition of miRNA-877-3p increased the cells’ motility (Figure 5D and Appendix A). Furthermore, Annexin V staining revealed that the miR-877-3p mimics could induce the apoptosis of PANC-1, SW1990 and MIA PaCa-2 cells (Figure 5E and Appendix A).

### 2.5. miRNA-877-3p Negatively Regulates Bcl-2 and Promotes Apoptotic Protein Expression

To further investigate the molecular mechanisms of miR-877-3p-modulated promotion of PANC-1 cells apoptosis, the dual-luciferase reporter assay was used to detect the link between miRNA-877-3p and the anti-apoptotic protein Bcl-2. The result demonstrated that miRNA-877-3p overexpression significantly decreased the firefly luciferase reporter activity of the Bcl-2 in 293T cells. Data further showed that Bcl-2 expression was dramatically increased by miR-877-3p inhibitor, compared with that in the miRNA-NC group (Figure 6A).

Additionally, the immunoblotting assay was used to detect the connection between miRNA-877-3p and apoptotic proteins, and the results demonstrated that miR-877-3p could promote the expression of apoptotic proteins (caspase 3, caspase 8, cleaved-caspase 8, cleaved-caspase 9 and Bax) and inhibit the expression of the anti-apoptotic protein Bcl-2, whereas the miRNA-877-3p inhibitor group showed the opposite results (Figure 6B). The above results showed that miRNA-877-3p could target Bcl-2 mRNA to regulate its expression, which was consistent with the previous report [29].

### 2.6. Rutin Affects the Behavior of Pancreatic Cancer Cells by Upregulating the Expression of miRNA-877-3p

To explore whether the effect of rutin on PANC-1 cells was mediated by upregulating miRNA-877-3p, we treated PANC-1 cells with both miRNA-877-3p inhibitor and rutin. The results of CCK-8, EdU, and scratch test assays showed that the inhibitory effects of rutin on the proliferation and migration of PANC-1 cells were weakened when treated with miR-877-3p inhibitor (Figure 7A–C). Similarly, miR-877-3p inhibitor reversed rutin’s inhibitory effect on SW1990 and MIA PaCa-2 cells’ proliferation (Appendix A). Our data also showed that rutin inhibited SW1990 cells’ migration by upregulating miR-877-3p expression (Appendix A). Additionally, the apoptosis analysis by FACS uncovered that the downregulation of miRNA-877-3p reversed the proapoptotic effect of rutin in PANC-1, SW1990 and MIA PaCa-2 (Figure 7D and Appendix A). An immunoblotting assay revealed that miRNA-877-3p inhibition blocked the rutin-induced cell apoptosis (Figure 7E). The above data show that rutin can regulate the apoptosis of PANC-1 cells through miR-877-3p.

## 3. Discussion

Increasingly, evidence shows that natural products, especially flavonoids, play a promising role in cancer prevention and treatment. These compounds act as antitumor agents by inhibiting proliferation, angiogenesis and metastasis, or inducing cell death [30]. Multiple studies have shown that rutin can act as an effective anti-tumor candidate [31]. Yang et al. first reported that rutin works as an antitumor factor, inhibiting tumor growth in human leukemia HL-60 cell in a xenograft animal model [32]. Other studies have also indicated that rutin can induce cell apoptosis by arresting the cell cycle or activating apoptosis-related pathways in colon [33] and breast cancers [34]. In addition, rutin reduced the phosphorylation of ERK1/2 to induce cell cycle arrest in the G2 phase and to inhibit proliferation [35]. However, the role of rutin in pancreatic cancer has been poorly studied. In our study, we revealed that rutin effectively impacted the cell fate of pancreatic cancer cells. Upon rutin treatment, cell proliferation and migration were repressed in pancreatic cancer cells. In addition, rutin was found to promote the apoptosis of pancreatic cancer cells by upregulating the expression of apoptotic proteins, i.e., Bax, cleaved caspase 3/8/9, and by downregulating the expression of the anti-apoptotic protein Bcl-2.

Accumulated evidence indicates that miRNA can be involved in the initiation and progression of cancer in several ways, such as by regulating the proliferation, differentiation, apoptosis, and development of cancer cells [36,37]. Several miRNAs have been shown to serve as tumor suppressors or oncogenes [38,39], and to provide new strategies for cancer diagnosis and treatment. The modulation of miRNAs expression by flavonoids has been studied in various human diseases, such as cancer and diabetes. For example, quercetin was identified as an anti-tumor treatment that works by suppressing the miR-21-PDCD4 signaling pathway and by inhibiting oxidative stress [40]. Resveratrol can modulate the expression of miR-17 in prostate cancer [41], and of miR-21 in pancreatic cancer [42] and prostate cancer [43]. Intriguingly, we observed that rutin could regulate miRNAs expression in PANC-1 cells, especially the increasing expression of miR-877-3p, which has been reported to play an important role in anti-tumor treatments.

miR-877-3p has been investigated in various diseases, e.g., miR-877-3p suppressed cell proliferation in bladder cancer by increasing the expression of the tumor suppressor gene p16 [26]; miR-877-3p was also discovered to be a suppressor for pulmonary fibrosis [44]. Another study demonstrated that the combination of miR-877 and miR-3619 could repress the invasion ability of breast cancer cells through inhibiting the activation of phospholipase D (PLD) [45]. However, our study unexpectedly found that miRNA-877-3p was significantly upregulated after rutin was administered by high-throughput sequencing. Furthermore, our results showed that miR-877-3p could inhibit the proliferation and migration and promote the apoptosis of pancreatic cancer cells, which could be blocked by miR-877-3p inhibitor. Interestingly, we also found that miR-877-3p downregulated Bcl-2 expression by directly targeting its 3′ UTR, which is consistent with previous studies [29]. Based on this, we propose the hypothesis that the effect of rutin on pancreatic cancer cells’ behaviors may be mediated by miR-877-3p. So rutin and miR-877-3p inhibitor were used to treat PANC-1 cells. The results show that the inhibitory effects of rutin on pancreatic cancer cells’ proliferation and migration and the effect of promoting cell apoptosis were attenuated by miR-877-3p inhibitor. Therefore, rutin regulated the expression of miR-877-3p to control the cell fate of pancreatic cancer cells, which might provide a new strategy for pancreatic cancer therapy. The sequencing results also showed that some other miRNAs in PANC-1 cells were upregulated by rutin, such as miR-451a and miR-302a-5p. It has been shown that miR-451a could induce tumor cell apoptosis and inhibit tumor cell growth in hepatocellular carcinoma [46]. In addition, miR-302a-5p/367-3p mediated the ability of HMGA2 to regulate the malignant behavior of endometrial carcinoma cells [47]. We wonder whether miR-451a, miR-302a-5p and/or miR-302a-5p/367-3p are involved in rutin-regulated PANC-1 death; this needs further investigation during our more comprehensive exploration of miRNAs in PC treatment in the future.

In conclusion, our results demonstrated that rutin played an inhibitory role in pancreatic cancer cells through mediating the expression of miR-877-3p. miR-877-3p targeted the Bcl-2 mRNA to suppress the expression of Bcl-2 to promote cell apoptosis, and our study also found that miR-877-3p could inhibit the proliferation and migration of pancreatic cancer cells. Therefore, this study is helpful in promoting the application of natural compound rutin as an adjuvant therapy for pancreatic cancer, and in providing a new possibility for the treatment of pancreatic cancer.

## 4. Materials and Methods

### 4.1. Rutin Preparation and Analysis

The rutin used in this study was obtained by ethanol extraction, as previously described [48]. In brief, 40 g of Tartary buckwheat seed powder (Dongfang Shengu Co., Ltd., Bijie, China) was mixed with 65% of ethanol at a material-to-solvent ratio of 1:10. This was heated at 75 °C for 4 h, and the mixture was sonicated. The extract was obtained by vacuum filtration. The extracted solution was concentrated by rotary evaporation at 50 °C until no alcohol remained. The extract was then purified using a chromatography column filled with AB-8 macroporous resin. The rutin was eluted with absolute ethanol solvent. The elution was further concentrated by rotary evaporation and lyophilized. The obtained Tartary buckwheat rutin was then analyzed using HPLC, and the purchased rutin with HPLC purity ≥ 98% (Solarbio, Beijing, China, CAS:153-18-4) was used as standard. The prepared rutin sample was stored at −80 °C and used for the following experiments.

### 4.2. Cell Cultures

A human pancreatic cancer cell line, PANC-1, was obtained from the Medical School of Fudan University in Shanghai, and another two pancreatic cancer cell lines, the SW1990 cell and MIA PaCa-2 cell, were obtained from Zhejiang University. All cells were subsequently cultured in DMEM (Gibco, Thermo Fisher, Shanghai, China) and supplemented with 10% Fetal Bovine Serum (FBS) (Gibco, Vienna, Austria) and 1% penicillin-streptomycin solution (Gibco, Thermo Fisher, Waltham, MA, USA). The cells were incubated at 37 °C under a humidified atmosphere with 5% CO_2_ and treated with rutin at indicated concentrations.

### 4.3. Cell Transfection

miRNA-877-3p mimics (miR-mimics), 5′-TCCTCTTCTCCCTCCTCCCAG-3′; miR-877-3p inhibitor (miR-inhibitor), 5′-CTGGGAGGAGGGAGAAGAGGA-3′; and their corresponding control oligonucleotides (miR-NC), 5′-GGCUCUAGAAAAGCCUA

UGC-3′, were synthesized by Ribo Bio (Guangzhou, China). Transfection was carried out with a final concentration of 50 nM miR-877-3p mimics and 50 nM miR-877-3p inhibitor using the Lipofectamine 2000 reagent (Invitrogen, Shanghai, China), following the manufacturer’s protocol.

### 4.4. RNA Isolution, Reverse Transcription, and qRT-PCR

After the cells were treated for 48 h, total RNA samples were extracted from the cells using Trizol reagent (Beyotime Shanghai, China). According to the manufacturer’s instructions, the RNA concentration and purity were measured by a NanoDrop spectrophotometer (NanoDrop 2000; Thermo Fisher, Waltham, MA, USA). Single-stranded complementary DNA (cDNA) was generated using a RIBO SCRIPTTM Reverse Transcription Kit (Ribo Bio, Hangzhou, China), following the manufacturer’s directions. The expression level of miRNA-877-3p was tested using SYBR Green (Ribo Bio, Hangzhou, China) through a LightCycler 480 II (Roche, Basel, Switerland). U6 served as an internal control of miRNAs. The relative quantitative expression was determined using the 2-ΔΔCT method. The primers were synthesized by Ribo (Ribo Bio, Hangzhou, China). The primer sequences were as follows: miR-877-3p: 5′-ATTCCTCTTCTCCCTCCTCCC-3′ (forward) and 5′-CAGTCAGGGTCCGAGGTAT-3′ (reverse); miR-NC: 5′-UACA ACCUCCUAGAAGAGUAGA-3′ (forward) and 5′-UCUACUCUUUCUAGGAGGUUG UGA-3′ (reverse); U6: 5′-TGCGGGTGCTCGCTTCGGCAGC-3′ (forward) and 5′-CAGT GCAGGGTCCGAGGTAT-3′ (reverse).

### 4.5. CCK-8 Assay

Cell viability was determined using CCK-8 (Cell Counting Kit-8), according to the manufacturer’s instructions. Approximately 2 × 10^3^ cells in the logarithmic growth phase were collected and then seeded into 96-well plates. Then, 10 μL CCK-8 reagents (Dojindo Laboratories, Kumamoto, Japan) were added to each well at 24 h and 48 h after the cells were transfected or treated with 5 μg/mL, 10 μg/mL, 20 μg/mL, or 40 μg/mL of rutin. The cells were incubated for 2 h, and a microplate reader (BioTek Instrument, Inc., Winooski, VT, USA) was then used to measure the optical density at a wavelength of 450 nm. The data of the OD450 value were used to reflect cell proliferation rates.

### 4.6. EDU Incorporation Assay

An EdU assay was conducted using a Cell-Light EdU DNA Cell Proliferation Kit (RiboBio, Guangzhou, China). Briefly, cells were first treated with rutin or transfection for 48 h and then incubated with EdU for 2 h. PANC-1 cells were fixed with 4% paraformaldehyde, stained with Apollo Dye Solution and then mounted with Hoechst 33342. The positive cells were photographed and counted using an Inverted fluorescence microscope (Leica, Wetzlar, Germany). 

### 4.7. Wound-Healing Assay

The ability of cell migration was evaluated using a wound-healing test. The cells grew to 80–90% confluence in 6-well plates, and then cell monolayers were scratched with a sterile 200 μL pipette tip across the center of the well to generate a clean, straight wound area. Rutin (5 μg/mL, 10 μg/mL) was added to the cells, or the cells were transfected with miR-877-3p mimics, miR-8773p inhibitor and their corresponding control oligonucleotides (miR-NC), and then the cells were cultured in the medium, supplemented with FBS-free. The morphology of the cells was observed at 0 h, 24 h and 48 h by light microscopy.

### 4.8. TUNEL Assay

A terminal-deoxynucleotidyl transferase-mediated Nick End Labeling (TUNEL) assay (Beyotime, Shanghai, China) was performed to determine the apoptotic cells, according to the manufacture’s instructions. Briefly, the cells were washed with PBS 48 h after administration or transfection, and 4% paraformaldehyde was fixed for 30 min and stained with TUNEL solution. Photographs were taken and analyzed under an inverted fluorescence microscope (Leica, Wetzlar, Germany).

### 4.9. Apoptosis Analysis

Cell apoptosis was analyzed with Annexin V-fluorescein isothiocyanate/propidium iodide (PI) double staining, using an Annexin V-FITC Apoptosis Detection Kit (Beyotime, Shanghai, China) according to the manufacture’s recommendations. Briefly, forty-eight hours after transfection or administration, 5 × 10^5^ cells were collected and washed with cold phosphate-buffered saline (PBS) twice. The collected cells were stained with Annexin V-FITC and PI in a binding buffer for 10 min. The stained cells were then analyzed using a BD AccuriTM C6 Plus Flow Cytometer (COE, BD Biosciences, Beijing, China).

### 4.10. Dual-Luciferase Reporter Assay

For the luciferase reporter assay, the 3′UTR of Bcl-2 was amplified and inserted into the pmirGLO luciferase vector (GeneCreat, Wuhan, China). The Bcl-2 (Ribo Bio, Hangzhou, China) was transfected with miR-877-3p mimic or miR-877-3p inhibitor into 293T cells, respectively, using lipo2000 (Invitrogen, Carlsbad, CA, USA), following the manufacturer’s procedures. Forty-eight hours after transfection, the cells were harvested, and luciferase activity was detected using the dual-luciferase reporter assay system. Renilla luciferase served as the internal control.

### 4.11. Western Blot Assay

Forty-eight hours after transfection or rutin administration, total proteins were extracted from PANC-1 cells with the radio immunoprecipitation assay (RIPA) lysis buffer (Beyotime, Shanghai, China). The supernatant was collected, and protein concentrations were determined using a BCA Protein Assay Kit (Beyotime, Shanghai, China). Proteins were separated by SDS-PAGE and subsequently transferred onto the polyvinylidene difluoride membranes (Millipore, Bedford, MA, USA) and blocked using 5% non-fat milk at room temperature for 2 h. The membranes were incubated with the primary antibody for anti-MMP9 (D361999, Sangon Biotech, Shanghai, China), anti-Bcl2 (ab182858, Abcam, Shanghai, China), anti-Bax (D120073, Sangon Biotech), anti-caspase-3 (AF1213, Beyotime, Shanghai, China), anti-caspase-8 (AF1243, Beyotime, Shanghai, China), anti-cleaved-caspase 9 (Asp353, Affinity, Liyang, China), anti-cleaved-caspase 3 (Asp175, Affinity, Liyang, China) and anti-β-actin (Beyotime, Shanghai, China) overnight at 4 °C. The membranes were then conjugated with a secondary anti-rabbit or anti-mouse (LI-COR, Lincoln, NE, USA) polyclonal antibody at room temperature for 1 h. The bands were visualized using the Enhanced Chemiluminescence System Reagent (KeyGENBioTECH, Nanjing, China).

### 4.12. Statistical Analysis

The experimental data were statistically analyzed by GraphPad Prism6.0 software and expressed as means ± standard error of measurement (SEM). Three replicates were set for each experiment, and an independent sample T-test was used for analysis, in which *p* < 0.05 was considered a significant difference (* *p* < 0.05, ** *p* < 0.01, *** *p* < 0.001, **** *p* < 0.0001), indicating that the data are statistically significant.

## Figures and Tables

**Figure 1 molecules-27-02293-f001:**
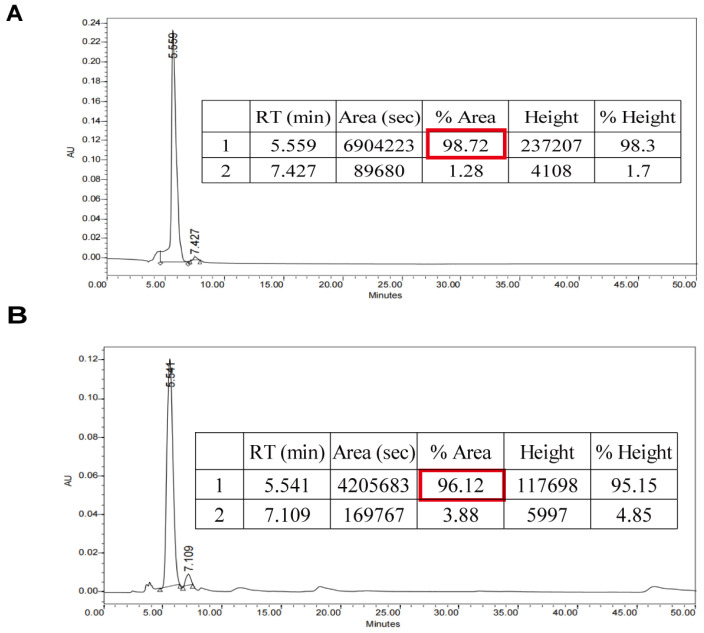
HPLC chromatograms of rutin. (**A**) Rutin standard, 98.72% purity. (**B**) Rutin from Tartary buckwheat, 96.12% purity.

**Figure 2 molecules-27-02293-f002:**
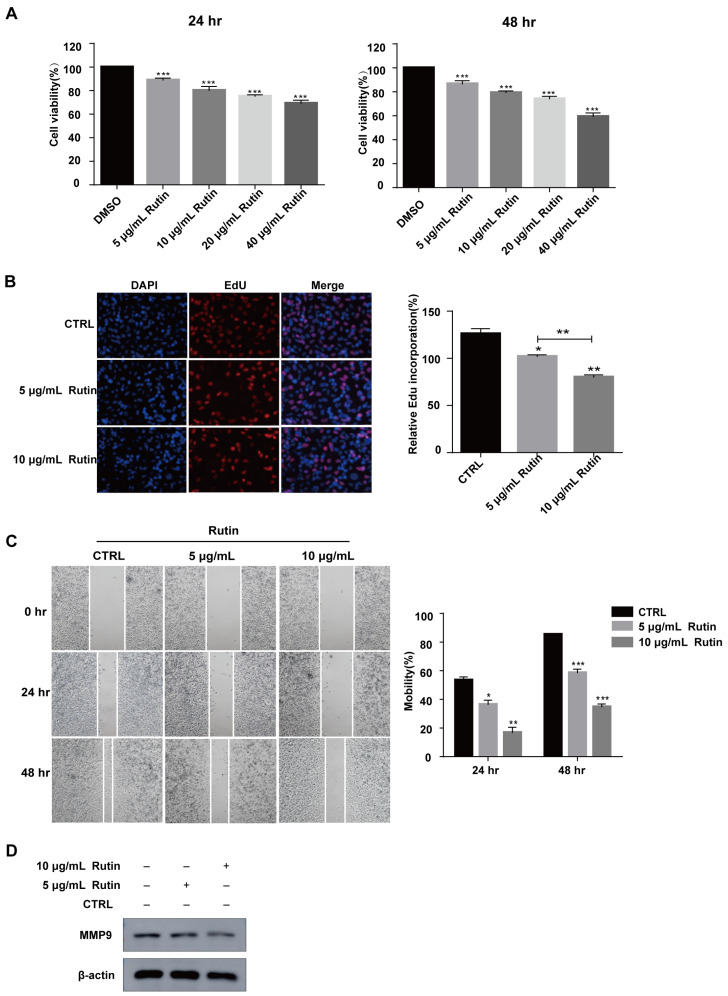
Rutin suppressed PANC-1 cell proliferation and migration in vivo. (**A**,**B**) CCK-8 and EdU assays revealed that PANC-1 cell proliferation was significantly suppressed after rutin treatment for 24 h or 48 h. (**C**) Wound-healing assay revealed that the migration ability of PANC-1 cells was significantly repressed after rutin treatment for 24 h or 48 h. (**D**) The expression of MMP-9 in PANC-1 cells was inhibited by rutin. * *p* < 0.05, ** *p* < 0.01, *** *p* < 0.001.

**Figure 3 molecules-27-02293-f003:**
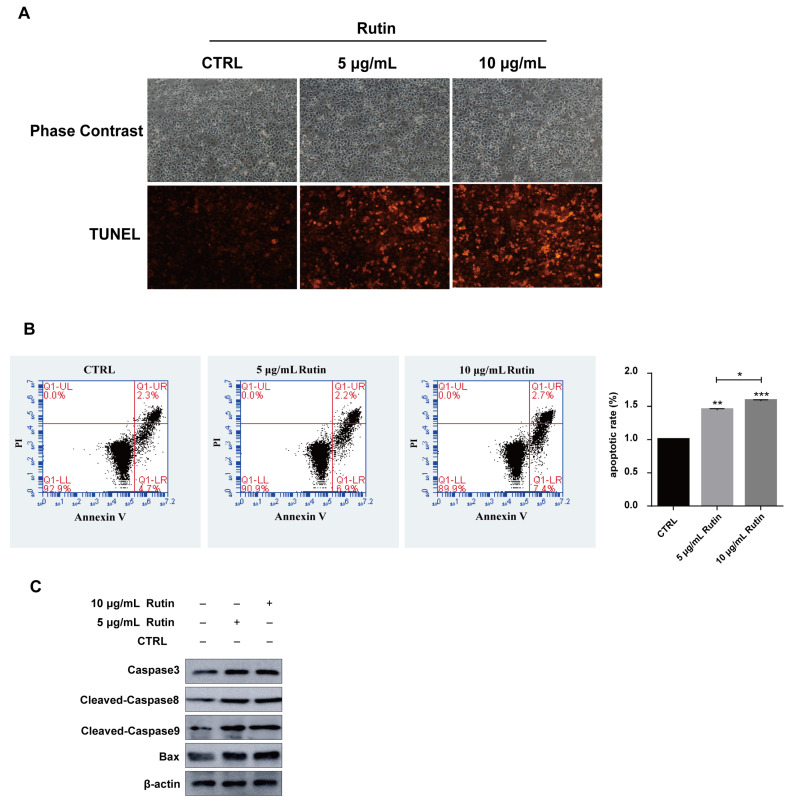
Rutin promoted PANC-1 cell apoptosis. (**A**) TUNEL assay revealed that rutin could promote PANC-1 cells apoptosis after rutin administration for 48 h. (**B**) Flow cytometric analysis also showed that PANC-1 cells apoptosis was enhanced by rutin administration for 48 h. (**C**) The expressions of caspase 3, cleaved-caspase 8, cleaved-caspase 9 and Bax were increased by rutin in the same treated PANC-1 cells. * *p* < 0.05, ** *p* < 0.01, *** *p* < 0.001.

**Figure 4 molecules-27-02293-f004:**
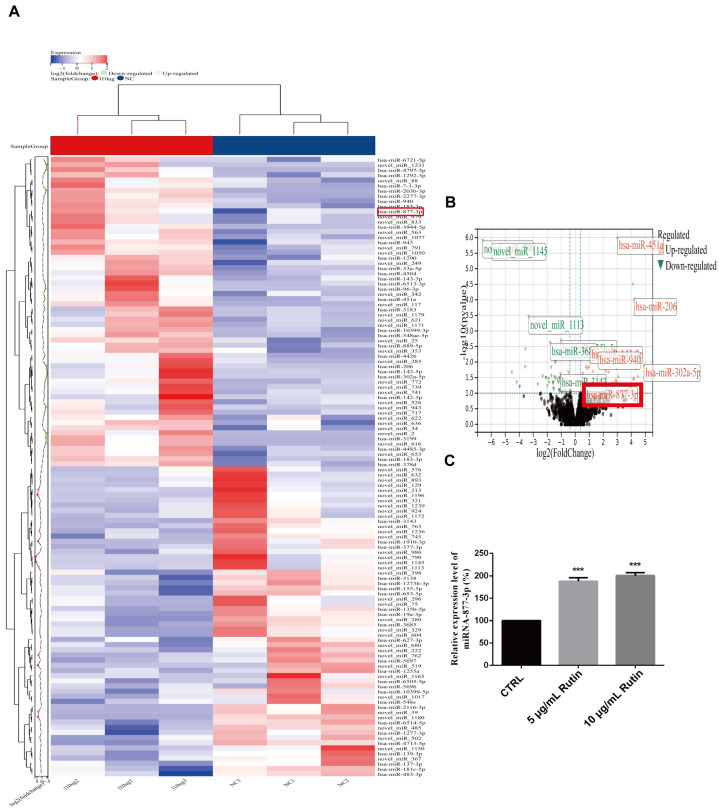
The effect of rutin on miRNA-877-3p expression. (**A**,**B**) High-throughput sequencing revealed that rutin upregulated the expression of miR-877-3p. (**C**) qRT-PCR analysis verified that rutin could significantly upregulate the expression of miRNA-877-3p. *** *p* < 0.001.

**Figure 5 molecules-27-02293-f005:**
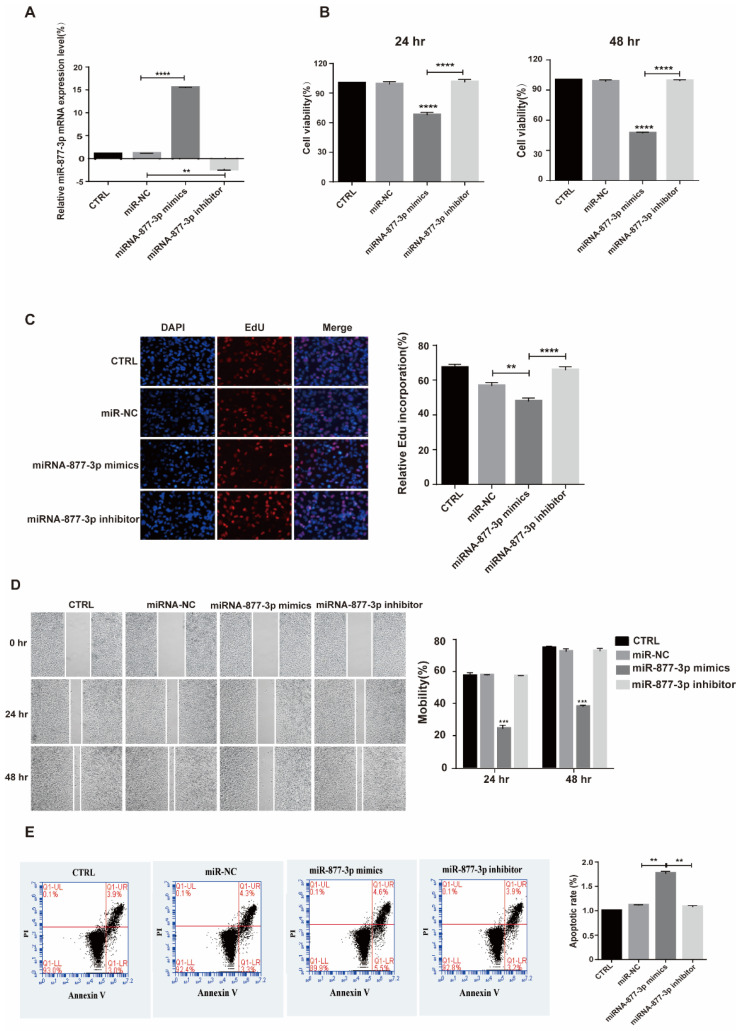
miRNA-877-3p suppressed PANC-1 cell proliferation and migration and promoted PANC-1 cell apoptosis. (**A**) miRNA-877-3p was significantly upregulated after transfection with miR-877-3p mimics but was downregulated after transfection with miR-877-3p inhibitor. (**B**,**C**) CCK-8 and EdU assays revealed that miRNA-877-3p mimics suppressed PANC-1 cell proliferation and miRNA-877-3p inhibitor promoted PANC-1 cell proliferation 24 h or 48 h after transfection. (**D**) Wound-healing assay showed that miRNA-877-3p mimics suppressed PANC-1 cell migration and miRNA-877-3p inhibitors promoted PANC-1 cell migration. (**E**) Flow cytometric assay showed that miR-877-3p mimics promoted PANC-1 cell apoptosis. ** *p* < 0.01, *** *p* < 0.001, **** *p* < 0.0001.

**Figure 6 molecules-27-02293-f006:**
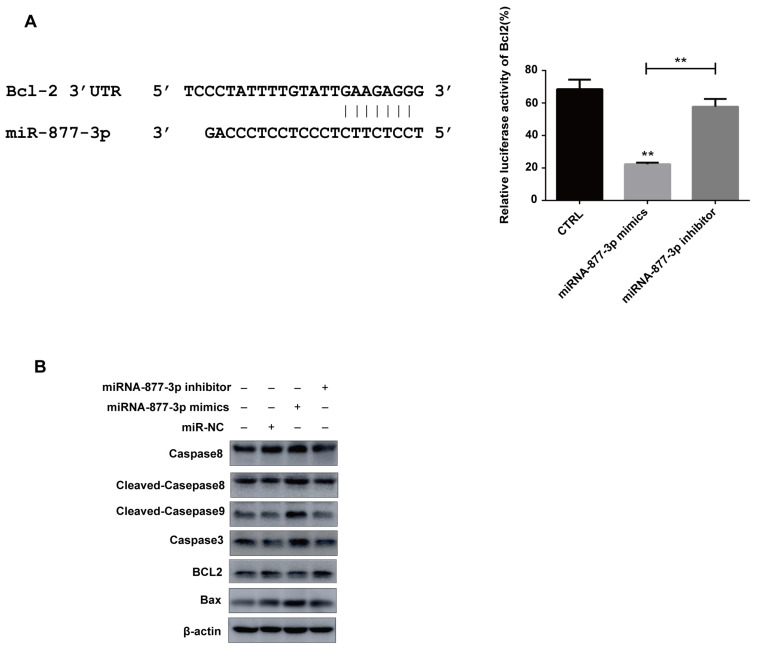
miRNA-877-3p negatively regulated Bcl-2. (**A**) Dual-luciferase reporter assay revealed that miR-877-3p overexpression significantly decreased the firefly luciferase reporter activity of the Bcl2. (**B**) miR-877-3p promoted the expression of apoptotic proteins caspase-8, cleaved-caspase 8, cleaved-caspase9, caspase-3 and Bax and inhibited anti-apoptotic protein Bcl2 expression. ** *p* < 0.01.

**Figure 7 molecules-27-02293-f007:**
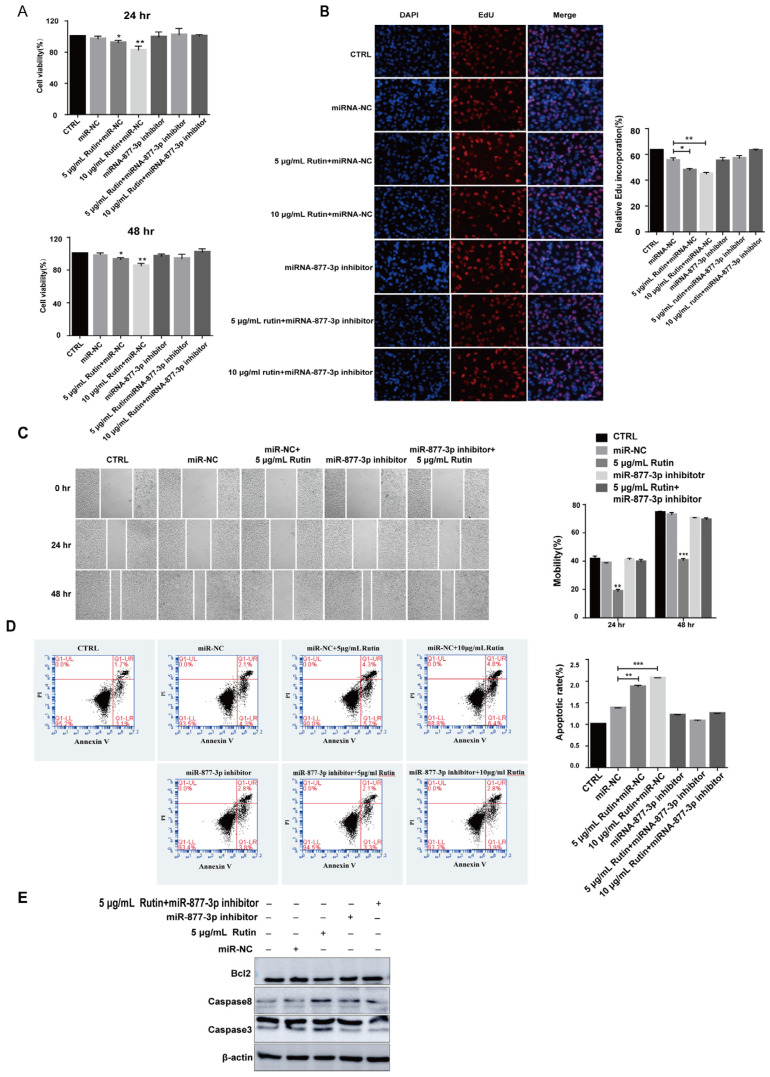
Rutin affected the behavior of PANC-1 cells by upregulating miR-877-3p expression. Downregulated miR-877-3p effectively reversed the effect of rutin on (**A**,**B**) proliferation, (**C**) migration and (**D**) apoptosis of PANC-1 cells. (**E**) Downregulation of miR-877-3p could reverse the effect of rutin on expression of the apoptotic proteins caspase 8, caspase 3 and anti-apoptotic protein Bcl2. * *p* < 0.05, ** *p* < 0.01, *** *p* < 0.001.

## Data Availability

The data presented in this study are available on request from the corresponding author.

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
