# Peer review of "Rutin Promotes Pancreatic Cancer Cell Apoptosis by Upregulating miRNA-877-3p Expression"

_molecules, 2022, doi:10.3390/molecules27072293_

Round 1

Reviewer 1 Report

In vitro model to study the antitumor effect of rutin on the human pancreatic cancer cell PANC-1 was described.

The work has well been performed but further details on the rutin and its chemical features and origin are required. On the contrary, a journal more medical-biological in nature could be better. 

Author Response

Point 1: The work has well been performed but further details on the rutin and its chemical features and origin are required. On the contrary, a journal more medical-biological in nature could be better. 

Response 1: Many thanks for your comment. The details on the rutin and its chemical features and origin were added in the revised manuscript.

Reviewer 2 Report

Dear Editor

Comments:

The manuscript entitled “Rutin promotes PANC-1 cell apoptosis by upregulating miRNA-877-3p expression” studies the effect of rutin on PAC- cell apoptosis via the regulation of miRNA-877-3p expression ntriguingly, they reported that rutin remarkably upregulated miR-877-3p expression to repress the transcription of Bcl-2 to induce cell apoptosis in PANC-1. However, there are some modifications required to be done before it is accepted for publication. The following are the specific comments on the manuscript:

Specific comments:

  1. Please add some numbers in the Abstract and Conclusion parts to make them more scientifically sound
  2. Additional information must be added to the Introduction section related to the therapeutic signaling pathways in cancer, the pharmacological properties of rutin, the mechanism of action of rutin
  3. Results: please make the figure more clear and organized.

Best regards

Author Response

Point 1: Please add some numbers in the Abstract and Conclusion parts to make them more scientifically sound

Response 1: Thanks for your comment. Appropriate numbers have been added in the corresponding sections in the revised manuscript.

Point 2: Additional information must be added to the Introduction section related to the therapeutic signaling pathways in cancer, the pharmacological properties of rutin, the mechanism of action of rutin.

Response 2: Thanks for your advice. Additional information about the therapeutic signaling pathways in cancer (Line 37-Line 41), the pharmacological properties of rutin (Line 57-Line 59), the mechanism of action of rutin (Line 59-Line 62) was added in the Introduction section in the revised manuscript according to your requirement.

Point 3: Results: please make the figure more clear and organized.

Response 3: Thanks for your advice. The figures were further modified and organised in the revised manuscript according to your requirement.

Reviewer 3 Report

In the present manuscript the authors aim to understand the mechanism of action of Rutin on pancreatic cancer cells. The authors hypothesize that Rutin promotes apoptosis in pancreatic cancer cells by up regulating the miRNA-877-3p expression. To address this the authors provided the data using a pancreatic cancer cell line PANC-1 and performed cell based experiments to conclude the findings. I believe that the authors needs additional experimental evidence to support their claim. Additional data and the following suggestion shall help improve the overall quality of the manuscript-

Major:

  1. The biggest concern about the manuscript is this one. All the experiment in the current manuscript are performed only on once pancreatic cancer cell line PANC-1. In order to reach to a concrete understanding of how rutin's mechanism of action works the authors need to  perform experiments using at least two more cell Ines to validate their findings.
  2. What was the rationale for authors to select only MMP9 and not other markers in figure 1D.
  3. The authors do not mention in legends, results and in methods how long the Rutin treatment was done on cells in figure 2C. This is consistent in many experiments. Kindly describe detailed figure legends.
  4. What was the rational the authors had to look for miRNA? Why was the screening performed only for miRNA? 
  5. The Figure 3B shows that it is not miRNA-877-3p but has-miR-451a is highly unregulated. why did the authors not validated it? and Why the miR-877 chosen for study? It is barely making the cutoff. 
  6. The membranes in wb for Casp8 in fig5a have double bands, why? it is not cropped well? Please update a clear image for Bcl2 in the same fig. The same needs to be done for fig.6E. 

Minor: 

  1.  The rutin that was used in majority of experiments needs to be mentioned in legends. I did not find in any methods section the source or purification of rutin. Please add the details in methods section.
  2. The abstract needs to be improvised in writing about the findings.
  3. The figure 1A please normalize the data with DMSO as internal control.
  4. Please highlight the miRNA -877-3p in figure 3A and C correctly. 
  5. What is TBF in figure 3C? update the legends.; also elaborate what NC in fig1A; Highlight mi877 in Fig 3A. 
  6. The primer seq info is missing in methods section. 

Author Response

Major:

Point 1: The biggest concern about the manuscript is this one. All the experiment in the current manuscript are performed only on once pancreatic cancer cell line PANC-1. In order to reach to a concrete understanding of how rutin's mechanism of action works the authors need to perform experiments using at least two more cell Ines to validate their findings.

Response 1: Thanks very much for the valuable advice. We are studying the function of rutin not only in pancreatic cancer cells, but also in colorectal cancer cells in our another research project. We found that rutin has similar functions for colorectal cancer cells, but based on different pathways. We will use more cell lines in the future to further study the exact mechanism of rutin in pancreatic cancer cells.

Point 2: What was the rationale for authors to select only MMP9 and not other markers in figure 1D.

Response 2: Thanks for your comment. Previous research showed that Matrix metalloproteinases (MMPs) have long been implicated for roles in cancer initiation, tumor growth, and metastasis (doi:10.1007/BF02574483.). MMP-9 is expressed by pancreatic cancer tumors, and MMP-9 have been suggested as biomarkers for pancreatic cancer (doi: 10.1097/MPA.0b013e3181a488a0.). Moreover, the pancreatic cancer cell line PANC-1 was found to produce MMP-9 (doi: 10.1016/bs.pmbts.2017.03.004.). So we selected MMP9 as a marker based on the literatures.

Point 3: The authors do not mention in legends, results and in methods how long the Rutin treatment was done on cells in figure 2C. This is consistent in many experiments. Kindly describe detailed figure legends.

Response 3: Thanks for your advice. The rutin treatment time has been added in the method. And the figure legends were further detailed in the revised manuscript according to your requirement.

Point 4: What was the rational the authors had to look for miRNA? Why was the screening performed only for miRNA? 

Response 4: MiRNA, as a small molecule RNA, has been studied more and more widely in the treatment of diseases. We pay attention to the miRNA changes mediated by rutin. Therefore, the miRNA expression under the action of rutin was analyzed by high-throughput sequencing. Based on the genes we are concerned about, we also verified the expression of the target miRNA by qPCR.

Point 5: The Figure 3B shows that it is not miRNA-877-3p but has-miR-451a is highly unregulated. why did the authors not validated it? and Why the miR-877 chosen for study? It is barely making the cutoff. 

Response 5: The inhibitory effect of mir-451a in many tumors has been described and reported. We expect to screen the potential miRNAs with new functions under the action of rutin. Although the expression rise of mir-877 is close to the threshold based on sequencing, the expression rise of mir-877 is more than 2-fold based on qPCR verification, and the repeatability is very good. Rutin mediated miRNA expression is multifaceted, and other miRNA molecular functions are also of our concern in the future.

Point 6: The membranes in wb for Casp8 in fig5a have double bands, why? it is not cropped well? Please update a clear image for Bcl2 in the same fig. The same needs to be done for fig.6E. 

Response 6: For CASP8, we consulted the antibody company and consulted the published articles. It is possible that there is another band above it. Relatively more clear images for Bcl2 were used in the Figures in the revised manuscript.

Minor:

Point 1:  The rutin that was used in majority of experiments needs to be mentioned in legends. I did not find in any methods section the source or purification of rutin. Please add the details in methods section.

Response 1: Many thanks.  The source or purification of rutin from tartary buckwheat was added in the Methods and materials Section (Line 280-Line 291, and Fig. 1).

Point 2: The abstract needs to be improvised in writing about the findings.

Response 2: Thanks for your advice. The result part was improvised in writing about the findings in the Abstract.

 Point 3: The figure 1A please normalize the data with DMSO as internal control.

Response 3: Thanks for your advice. We normalized the data with DMSO as internal control in figure 1A in the revised manuscript according to your requirement.

Point 4: Please highlight the miRNA -877-3p in figure 3A and C correctly. 

Response 4: Thanks for your advice. MiR-877-3p has been highlighted in figure 3A and B correctly in the revised manuscript.

Point 5: What is TBF in figure 3C? update the legends.; also elaborate what NC in fig1A; Highlight mi877 in Fig 3A. 

Response 5: Thanks for your advice. In the revised manuscript, we corrected the wrong writing of TBF into rutin, and correctly highlighted miR-877-3p in red color in figure 3A . Rutin was dissolved in DMSO, so DMSO was used as a negative control Fig1A.

Point 6: The primer seq info is missing in methods section. 

Response 6:  Many thanks. The primer sequences “( miR-877-3p: 5’-ATTCCTC

TTCTCCCTCCTCCC-3’ (forward) and 5’-CAGTCAGGGTCCGAGGTAT-3’ (reverse); miR-NC: 5’-UACAACCUCCUAGAAGAGUA GA-3’ (forward) and 5’-UCUACUCUU

UCUAGGAGGUUGUGA-3’ (reverse); U6: 5’-TGCGGGTGCTCGCTTCGGCAGC-3’ (forward) and 5’-CAGTGC AGGGTCCGAGGTAT-3’ (reverse)” have been added in methods section in the revised manuscript.

Reviewer 4 Report

In this manuscript, Liao and coworkers reported a molecular study on the response of rutin in the human pancreatic cancer cell line PANC-1. The authors validated the cancer repression function of the natural flavonoid rutin and identified the miRNA-877-3p as one downstream effector of rutin. The authors further proved the tumor repression function of rutin and miRNA-877-3p is dependent on Bcl-2-based cell apoptosis pathway by various epistasis analyses.

The documented data support the conclusion well and show novel insight into the molecular mechanism of rutin in cancer biology. Therefore, this work meets the requirements of originality and scope by the journal Molecules. The experiments in this manuscript are straightforward and well documented. Only few minor points need to be addressed before publication in Molecules.

In Figure 2B, the authors showed two dimensions of cell staining (PE channel and FITC channel) in the flow cytometry data but only gave the staining information of Annexin V (FITC channel) in the text. What is the target of the other channel?

For all the flow cytometry data, it is more informative to readers if label the axes by staining/target rather than fluorophore/color.

miR-877-3p is hard to find in current Figure 3A and 3B, which should be highlighted. A table summarizing miRNAs that are up or down regulated from the RNA-seq analysis is suggested.

The miR-877-3p is not the miRNA with highest increase in expression. Other miRNAs with more significant fold changes should be discussed to give readers a more comprehensive view. For one instance, the miR-451a induces apoptosis and suppress tumor growth (doi: 10.33594/000000118), so does miR-302a-5p (doi: 10.1186/s13046-018-0686-6) and few other miRNAs from the Figure 3B.

The oligonucleotides sequences of miRNA-877-3p mimics (miR-mimic), miR-877-3p inhibitors (miR-inhibitor), and control oligonucleotides (miR-NC) should be given in the Material and Method section.

Author Response

Point 1: In Figure 2B, the authors showed two dimensions of cell staining (PE channel and FITC channel) in the flow cytometry data but only gave the staining information of Annexin V (FITC channel) in the text. What is the target of the other channel? 

Response 1: PE channel detects PI staining and FITC channel detects annexin V staining. Such double staining method can more accurately analyze apoptotic cells.

Point 2: For all the flow cytometry data, it is more informative to readers if label the axes by staining/target rather than fluorophore/color.

Response 2:  Thanks for your advice. We corrected our previous label and labelled the axes by staining target in all the flow cytometry figures in the revised manuscript.  

Point 3: miR-877-3p is hard to find in current Figure 3A and 3B, which should be highlighted. A table summarizing miRNAs that are up or down regulated from the RNA-seq analysis is suggested.

Response 3:  Thanks for your advice. MiR-877-3p has been highlighted in figure 3A and B correctly in the revised manuscript. Because sequencing showed that there were too many miRNAs differentially expressed under the influence of rutin, we thought it might be more intuitive to use heat map.

Point 4: The miR-877-3p is not the miRNA with highest increase in expression. Other miRNAs with more significant fold changes should be discussed to give readers a more comprehensive view. For one instance, the miR-451a induces apoptosis and suppress tumor growth (doi: 10.33594/000000118), so does miR-302a-5p (doi: 10.1186/s13046-018-0686-6) and few other miRNAs from the Figure 3B.

Response 4:  Thanks for your advice. In this study,we are more concerned about the less reported miRNA in functional reports, hoping to explain more comprehensively the role of Rutin in inhibiting pancreatic cancer cells. MiR-451a has been well reported in the inhibition of tumors and even pancreatic cancer. We think that the function of miR-451a only under rutin may not be sufficient. The expression and functional verification based on mir-877-3p also proved our hypothesis. Rutin-mediated miRNA expression is multifaceted, and other miRNA molecular functions are also of our concern.

Point 5: The oligonucleotides sequences of miRNA-877-3p mimics (miR-mimic), miR-877-3p inhibitors (miR-inhibitor), and control oligonucleotides (miR-NC) should be given in the Material and Method section.

Response 5: Thanks very much. The oligonucleotides sequences of miRNA-877-3p mimics (5’-TCCTCTTCTCCCTCCTCCCAG-3’), miR-877-3p inhibitors (5’-CTGG

GAGGAGGGAGAAGAGGA-3’), and control oligonucleotides (5’- GGCUCUAGAA

AAGCCUAUGC-3’) were provided in the Material and Method section in the revised manuscript.

Round 2

Reviewer 2 Report

Good luck

Author Response

Dear reviewer,

Thank you very much for your comments.

Reviewer 3 Report

In the present manuscript, the authors included majority of major and minor points which were crucial for improving the overall quality of the manuscript. However, the authors still did not show additional data from another pancreatic cell line which can support the basic hypothesis of the manuscript. All the experiments so far are performed in only one pancreatic cell line on the basis of which it would be inappropriate to conclude the findings of the manuscript. The authors claim to have observed similar effects of Rutin on other cancer cell lines but with different pathway but did not provide any of such data in supplements. I request the authors to at least repeat some of the key experiments from this manuscript in two pancreatic cell lines to support their hypothesis. 

I believe addressing this is highly crucial for the present manuscript. Thank you. 

Author Response

Point 1: In the present manuscript, the authors included majority of major and minor points which were crucial for improving the overall quality of the manuscript. However, the authors still did not show additional data from another pancreatic cell line which can support the basic hypothesis of the manuscript. All the experiments so far are performed in only one pancreatic cell line on the basis of which it would be inappropriate to conclude the findings of the manuscript. The authors claim to have observed similar effects of Rutin on other cancer cell lines but with different pathway but did not provide any of such data in supplements. I request the authors to at least repeat some of the key experiments from this manuscript in two pancreatic cell lines to support their hypothesis. 

Response 1: Thank you for your comment. We have successfully repeated some key experiments in another two pancreatic cancer cell lines SW1990 and MIA PaCa-2, including 1:Rutin affects the SW1990 and MIA PaCa-2 cells behavior (Figure S1); 2:miRNA-877-3p suppressed cells proliferation, migration and promoted cells apoptosis in SW1990 and MIA PaCa-2 cells (Figure S2); 3.Rutin affected the behavior of the cells by upregulating miR-877-3p expression in SW1990 and MIA PaCa-2 cells (Figure S3). Therefore, we think our data can support hypothesis of our manuscript. Thank you again for your reviewing.  
